# The “Double-Edged Sword” Effect of Perceived Algorithmic Control on Platform Workers’ Work Engagement

**DOI:** 10.3390/bs16010033

**Published:** 2025-12-23

**Authors:** Jian Zhu, Yuhe Cao, Yanjun Wang

**Affiliations:** Business School, Xiangtan University, Xiangtan 411105, China; 202421020280@smail.xtu.edu.cn (Y.C.); 13721452213@163.com (Y.W.)

**Keywords:** perceived algorithmic control, promotion-focused job crafting, prevention-focused job crafting, work engagement, algorithmic literacy

## Abstract

With the deep development and iterative upgrading of algorithmic technology, the management practice of platform enterprises using intelligent algorithmic technology has become a hot issue. However, there is little research on the impact of perceived algorithmic control on work engagement from the perspective of platform workers. Drawing upon the regulatory focus theory, this study constructs a “double-edged sword” model to test the impact of perceived algorithmic control on platform workers’ work engagement by focusing on the positive mediating role of promotion-focused job crafting, the negative mediating role of prevention-focused job crafting, and the moderating role of algorithmic literacy. The data collected from 302 platform workers in China were used for an empirical study, and corresponding analyses were carried out to verify the theoretical model constructed by using SPSS and Mplus. The findings indicate the following: (a) perceived algorithmic control positively affects work engagement through promotion-focused job crafting; (b) perceived algorithmic control negatively affects work engagement indirectly through prevention-focused job crafting; (c) the indirect effect of perceived algorithmic control on work engagement via promotion-focused job crafting is stronger when there is a high level of algorithmic literacy and weaker in the case of low algorithmic literacy; and (d) the indirect effect of perceived algorithmic control on work engagement via prevention-focused job crafting is weaker in situations of high algorithmic literacy and stronger in those of low algorithmic literacy. The findings not only enrich theoretical studies on algorithmic control and work engagement but also offer guidance to platform-based enterprises on how to leverage the positive aspects of algorithmic control to better support individuals with different traits.

## 1. Introduction

A large number of platform-based companies relying on internet technology have rapidly emerged in the context of the internet. These platform enterprises use advanced Internet technology to closely connect service demanders and suppliers, and workers can achieve rapid and precise matching with numerous job opportunities, which has led to the emergence of platform workers ([13]; [35]). China is one of the world’s largest and most dynamic digital platform markets. To date, the number of platform gig workers in China has exceeded 200 million. Among them, typical examples are platform gig workers such as food delivery riders and ride-hailing drivers who accept orders through platforms like Meituan and Didi. The market scale of China’s platform economy has grown rapidly, and the total number of workers in new forms of employment in China has also shown a growing trend. The rapid rise of the platform economy has driven the development of China’s economy.

Under the sharing economy, platform managers cannot implement on-site coordination and supervision, rendering traditional regulatory methods ineffective. Platform companies use algorithms for management and resource coordination, executing matching and control functions in a highly automated and data-driven manner to achieve organizational goals ([36]; [6]). Control is “the most fundamental problem of management”, and algorithmic control is at the core of operations in online labor platforms ([38]).

As an emerging management paradigm, algorithmic control has attracted significant scholarly attention ([29]), with research exploring its impact on work intentions, turnover behavior ([8]), perceived organizational support ([15]), employee well-being, psychological contracts ([41]), and service performance ([25]). However, the impact of perceived algorithmic control, which is defined as platform workers’ cognition and feelings about the algorithms that monitor their work, on work engagement has received relatively limited scholarly attention. Work engagement, defined as a positive, fulfilling, work-related state of mind characterized by vigor, dedication, and absorption ([2]), is crucial for organizational performance and competitive advantage. Given the distinct nature of labor relations under flexible employment and the reconstructed work forms in the platform context, findings from traditional employment settings are not fully applicable. Therefore, investigating how perceived algorithmic control affects platform workers’ work engagement is both timely and necessary. Nevertheless, the existing literature on the relationship between perceived algorithmic control and work engagement is limited, with related studies predominantly focusing on either positive or negative correlations from a singular perspective. Some studies suggest that algorithmic control can directly enhance work engagement ([17]) or indirectly promote it through gamification ([20]). In contrast, others contend that it undermines engagement by eroding work autonomy and increasing work complexity ([24]; [38]). This dichotomy suggests a potential “double-edged sword” effect, a metaphor acknowledged conceptually ([4]; [23]) but lacking a coherent theoretical explanation for its simultaneous opposing pathways.

To address this research gap, we draw on regulatory focus theory ([14]), which posits that individuals operate under two distinct self-regulatory systems: a promotion focus concerned with aspirations, accomplishments, and growth, and a prevention focus concerned with safety, responsibilities, and avoiding losses. We contend that perceived algorithmic control can simultaneously activate both systems, thereby inducing a “double-edged sword” effect on platform gig workers’ work engagement through divergent behavioral pathways. This theoretical framework allows us to integrate conflicting empirical findings into a cohesive dual-pathway model, moving beyond a simplistic positive/negative dichotomy.

As platform enterprises’ organizational structure and work design have become significantly decentralized ([39]), platform workers are not passive recipients of control but active agents who reshape their jobs—a process known as job crafting ([33]). This involves workers proactively changing their work in various ways, such as adjusting tasks, altering social interactions, and redefining their work’s meaning ([22]). Building on this conceptual foundation and aligning with regulatory focus theory, [19] ([19]) distinguished between promotion-focused job crafting (proactively seeking resources and challenges) and prevention-focused job crafting (reducing demands and avoiding risks). We propose these two crafting behaviors serve as the key mediating mechanisms that translate perceived algorithmic control into, respectively, higher or lower work engagement. Specifically, when platform workers perceive algorithmic control as providing enhanced job resources, opportunities, flexibility, and interest, it triggers a promotion focus, motivating them to adopt promotion-focused job crafting to maximize resource acquisition, thereby increasing work engagement. Conversely, when perceived algorithmic management manifests as negative effects such as work overload and job insecurity ([38]), it activates a prevention focus, leading workers to adopt prevention-focused job crafting ([5]), which consequently decreases work engagement.

Furthermore, a critical question remains: how can the positive path be amplified and the negative path mitigated? We introduce algorithmic literacy as a pivotal boundary condition. Grounded in the work of [11] ([11]), we conceptualize it as a cognitive competency comprising two core dimensions: awareness of algorithmic use across digital platforms and knowledge about how algorithms operate. We theorize that this literacy empowers workers by bridging the knowledge gap between them and the platform’s opaque structural controls. With higher literacy, workers can interact with the algorithmic system more self-determinedly, thereby fostering a promotion focus. Conversely, low literacy may exacerbate feelings of powerlessness and trigger a stronger prevention focus.

In summary, grounded in Regulatory Focus Theory, this study constructs a moderated mediation model to investigate the “double-edged sword” effect of perceived algorithmic control on platform workers’ work engagement. Its core research objective is to examine how perceived algorithmic control influences platform workers’ work engagement through promotion-focused and prevention-focused job crafting, as well as the moderating role of algorithmic literacy in the mediating pathways of these two types of job crafting. The study makes three key theoretical contributions. First, it advances beyond the prevailing yet simplistic “double-edged sword” metaphor by proposing a theoretically grounded dual-pathway model that explains how opposing effects coexist. Second, it elucidates the distinct mediating roles of promotion-focused and prevention-focused job crafting, thereby introducing a nuanced agency-centered mechanism that clarifies how perceived control translates into engagement outcomes. Third, it reframes algorithmic literacy not merely as an individual trait but as a critical competency that moderates these pathways, thereby bridging individual cognitive resources with the broader socio-technical dynamics of algorithmic management.

## 2. Theory and Hypotheses

### 2.1. Regulatory Focus Theory

Regulatory Focus Theory (RFT) examines how individuals approach positive goals and avoid negative ones. It posits two distinct self-regulatory systems governing human motivation: a promotion focus, concerned with aspirations and accomplishments, and a prevention focus, concerned with safety and responsibilities. Individuals with different regulatory foci exhibit distinct preferences in motivational pursuits and behavioral strategies. Building on this theory, scholars such as [19] ([19]) categorized the four-dimensional job crafting model proposed by [32] ([32]) into “promotion-focused job crafting” and “prevention-focused job crafting”. Individuals who engage in promotion-focused job crafting strive for self-development and greater work achievements. They proactively acquire and utilize various job resources, seize opportunities for personal growth, and undertake challenging tasks. In contrast, those who adopt prevention-focused job crafting aim to avoid failure and fulfill their safety needs. Consequently, they tend to exhibit prudent and conservative behaviors to ensure their own security ([21]; [18]; [42]).

According to the regulatory focus theory, individuals will have different strategic tendencies towards the same situation ([19]). Algorithmic control refers to the process in which the online labor platform uses algorithmic technology to monitor the behavior of platform workers and ensure that their behavior is consistent with the requirements of digital platform enterprises ([23]). Perceived algorithmic control refers to platform workers’ cognition and feelings about the algorithms used by the platform to monitor their work processes. For example, platform workers may perceive how the platform’s algorithms allocate tasks and evaluate work performance. When an individual perceives that the algorithm provides them with more job opportunities or receives positive work feedback, they may have a positive emotional experience and adopt promotion-focused job crafting. However, under the same algorithmic control, when an individual feels tense and anxious, which further leads to work pressure, they may adopt a conservative working style to reduce risks, resulting in prevention-focused job crafting. Therefore, we predict that the perceived algorithmic control of platform workers may lead to different job crafting behaviors, thus having a “double-edged sword” impact on work engagement.

### 2.2. The Mediating Role of Promotion-Focused Job Crafting in the Relationship Between Perceived Algorithmic Control and Work Engagement Among Platform Workers

When platform workers feel that algorithmic control helps them efficiently complete their tasks while enhancing work flexibility and challenge, they are more likely to adopt an open and accepting attitude towards the platform’s algorithmic control, thereby engaging in promotion-focused job crafting. Specifically, platform workers gradually benefit from the advanced technological support, rich informational resources, and real-time online guidance provided by algorithmic systems, effectively reducing distractions, inhibitory actions, and judgment activities during work. For instance, ride-hailing platforms’ intelligent matching functions cut down workers’ inefficient search costs, and real-time data feedback (e.g., delivery time alerts, service rating announcements) enables them to quickly calibrate work strategies; additionally, algorithm-designed flexible working hour mechanisms and tiered incentive rules (such as peak-hour subsidies and order-completion bonuses) not only improve the flexibility of work arrangements but also set progressive work challenges for workers, weakening their resistance to algorithms and fostering the perception of “algorithms empowering work”. For example, the company relies on various algorithms and driver applications for customer matching and dynamic pricing. Moreover, real-time voice traffic alerts reduce driver distraction and promote focus ([27]). Instant delivery platforms’ real-time voice navigation avoids safety hazards caused by riders checking mobile maps while on the move. Additionally, the platform’s algorithmic control proposes service requirements such as work hours and responsibilities based on the platform workers’ past performance data. These concrete and digitalized requirements deepen workers’ understanding, making the system a technological partner that better assists them in achieving their established goals. Faced with the control of algorithmic systems and the work demands they impose, platform workers are more likely to perceive these demands as challenging stress, which stimulates their intrinsic motivation, fosters stronger self-performance drive, and leads to more sustained behaviors ([31]).

Consistent with the core proposition of regulatory focus theory, when external contexts provide growth-oriented resources and challenging opportunities, individuals’ promotion-focused motivation is activated, driving them to proactively adjust work patterns to pursue higher goals ([14]). The resource support and challenge setting brought by perceived algorithmic control precisely meet this triggering condition, further motivating workers to engage in promotion-focused job crafting by expanding work boundaries, integrating resources, and optimizing processes. Thus, the perception of algorithmic control could lead to promotion-focused job crafting.

**H1a.** 
*Perceived algorithmic control positively associates with promotion-focused job crafting.*


Moreover, platform workers’ promotion-focused job crafting is positively correlated with work engagement. Promotion-focused individuals influence work engagement by reshaping three dimensions: structural work resources, social work resources, and challenging work demands ([31]). Work engagement refers to individuals’ physical, cognitive, and emotional integration with their work roles, encompassing vigor, dedication, and absorption ([16]). It is essentially a psychological state of positive individual-work context interaction, closely linked to workers’ proactive adjustment of work resources and demands—aligning with the core of promotion-focused job crafting, which emphasizes growth, achievement pursuit, and employees’ autonomy to optimize work.

First, structural work resource reshaping (e.g., seeking training, optimizing tools) enhance job competence. For instance, ride-hailing drivers learning communication skills via platform training or delivery riders upgrading smart equipment boost work control, thereby stimulating work vigor (sustaining energy in high-intensity work). Second, social work resource reshaping (e.g., connecting peers through algorithm-based communities, building customer relationships) expands social support networks, strengthening professional identity and fostering work dedication (willingness to invest extra effort and take pride in work). Third, challenging work demand reshaping (e.g., undertaking high-difficulty orders, participating in innovation pilots) satisfies achievement needs; breaking capacity boundaries through autonomous work adjustment generates self-actualization, improving work absorption (full immersion in work, immune to distractions) ([31]). Existing research confirms that autonomous work control and capacity improvement significantly enhance emotional and cognitive work investment ([2]).

**H1b.** 
*Promotion-focused job crafting positively associates with work engagement.*


Therefore, we believe that, in the course of work, platform workers perceive algorithmic control as providing more accessible job resources, greater career development opportunities, more challenging work demands, and more flexible work arrangements. This positive attitude toward algorithmic control allows them to view it as an opportunity, actively engaging in promotion-focused job crafting to acquire, coordinate, and leverage various work resources, thus achieving job performance and fulfilling their development needs, leading to positive work emotions and promoting greater work engagement. Accordingly, we propose the following hypotheses:

**H1c.** 
*Perceived algorithmic control positively associates with work engagement through promotion-focused job crafting.*


### 2.3. The Mediating Role of Prevention-Focused Job Crafting in the Relationship Between Perceived Algorithmic Control and Work Engagement Among Platform Workers

In the era of the platform gig economy, various platforms widely apply emerging algorithmic technologies to reduce management costs and improve service quality. The algorithmic control systems they establish include functions for algorithmic guidance, evaluation, and constraint management functions ([34]).

Platform workers’ perception of algorithmic control may have a positive predictive effect on prevention-focused job crafting. First, algorithmic guidance serves as the basis for platform workers’ decision-making. The algorithmic system sets role expectations and relevant job requirements, guiding or supporting them to complete tasks according to the platform’s established rules. Platform workers form their own understanding and judgment regarding platform standards and policies, which then influences their behavior. When platform workers perceive algorithmic norms and guidance and feel that the platform is pressuring them to improve efficiency and restricting their behavior, they may experience negative emotions such as job insecurity ([34]), leading them to avoid taking on excessive challenging demands and reducing hindrance demands. Second, tracking evaluation refers to the process where algorithms automatically record information about platform workers’ online labor and provide timely feedback and assessments of their service quality ([37]). Tracking evaluation enables platform workers to have a clearer understanding of their service context and quality, helping them improve their subsequent work ([37]). Despite platform workers’ reliance on algorithmic technical support and their continual self-improvement under algorithmic control, prolonged use may lead to the “paradox of autonomy bringing about a non-autonomous work state”. Additionally, platforms adopt a customer demand-oriented approach, where customers can monitor the service process in real-time through the algorithm and evaluate or rate the service at any time. These ratings ultimately impact the performance of platform workers, increasing their sense of uncertainty and frustration. Third, constraints management involves motivating platform workers through reward or punishment mechanisms, encouraging them to achieve predetermined goals and regulate their own behavior, while reducing inappropriate behaviors at work, such as working with negative emotions or charging additional fees, in order to avoid penalties ([4]). When the level of perceived algorithmic control is high, platform workers are more inclined to view the algorithmic system as controlling rather than informational. In order to accurately understand the dispatching mechanism and evaluation system of online labor platforms and achieve higher performance and actual earnings, platform workers are forced to comply with algorithmic control. Over time, this may reduce the work enthusiasm and initiative of platform workers, leading them to adopt prevention- focused job crafting.

**H2a.** 
*Perceived algorithmic control positively associates with prevention-focused job crafting.*


The adoption of prevention-focused job crafting by platform workers exerts a significant negative impact on work engagement. Work engagement is defined as a positive and fulfilling work-related psychological state ([3]), and this negative association stems from a fundamental motivational conflict: specifically, prevention-focused job crafting is centered on loss avoidance ([14]), whereas work engagement relies on individuals’ intrinsic drive to proactively pursue growth and achievement. When engaging in prevention-focused job crafting, platform workers tend to adopt algorithm avoidance behaviors (e.g., ignoring in-app reminders, evading real-time algorithmic monitoring) to mitigate work risks. While such actions can reduce short-term errors, they cut off the critical channel for workers to improve their skills through algorithmic feedback, thereby undermining their work self-efficacy and work vigor. Crucially, platform work demands do not diminish with workers’ defensive postures; on the contrary, they accumulate continuously as business operations advance, widening the gap between workers’ defensive performance and platform standards. This discrepancy triggers intense feelings of helplessness and chronic psychological stress, which further induce occupational burnout and lead to a marked decline in workers’ overall level of work engagement.

**H2b.** 
*Prevention-focused job crafting negatively associates with work engagement.*


Based on the above analysis, this study suggests that the pressure and burden brought by perceived algorithmic control in platform gig work pose a threat to both physical and mental health, as well as to work performance. As a result, workers choose to adopt prevention-focused job crafting, avoiding challenges at work, becoming more passive and disengaged, which ultimately reduces work engagement. Thus, the following hypotheses are proposed:

**H2c.** 
*Perceived algorithmic control negatively associates with work engagement through prevention-focused job crafting.*


### 2.4. The Moderating Role of Algorithmic Literacy

Algorithmic literacy is defined as the awareness of algorithms’ extensive application on online platforms, an understanding of their operational mechanisms, the ability to critically assess their impacts, and the capacity to formulate appropriate responses—all while being able to coexist with algorithms ([11]). It thus enables individuals to not only recognize and understand algorithmic operations in digital services but also to better adapt to an algorithmic society ([1]).

Different levels of algorithmic literacy affect individuals’ cognition, attitudes, and behaviors toward algorithmic control in the workplace. When platform workers possess a higher level of algorithmic literacy, they tend to have a thorough understanding of algorithms and are better able to comprehend how algorithmic control operates at different stages. They also have the skills to respond to, and even influence, algorithmic operations. The combination of these cognitive and behavioral dimensions enables platform workers to understand, assess, and respond to algorithms in a self-determined way, aligning the algorithmic system with their own needs and expectations based on subjective requirements. In practice, this means individuals are able to apply strategies that allow them to modify predefined settings within the algorithm-driven environment ([11]), aligning it more closely with their expectations and needs. In doing so, this not only aids and encourages them to efficiently accomplish tasks assigned by the platform but also elevates their self-efficacy perceptions and fosters increased work engagement ([7]). Therefore, a high level of algorithmic literacy encourages platform workers to adopt proactive strategies aimed at achieving positive outcomes, meaning they are more likely to engage in promotion-focused job crafting rather than resort to defensively oriented job crafting strategies.

**H3a.** 
*Algorithmic literacy positively moderates the relationship between perceived algorithmic control and promotion-focused job crafting.*


Similarly, according to regulatory focus theory, platform workers with low levels of algorithmic literacy have a less comprehensive understanding of how algorithmic systems operate. This lack of understanding can lead to challenges at work, making them more likely to adopt conservative behavioral strategies in response to algorithmic control, engaging in prevention-focused job crafting to avoid penalties and minimize personal losses. When the level of algorithmic literacy is relatively low, platform workers possess limited algorithmic knowledge and lack understanding of how algorithmic systems operate and how information is disseminated within these systems ([12]). Due to their vague understanding of platform rules and algorithmic mechanisms, they are forced to passively accept decisions made under algorithmic control, leading to a high degree of distrust toward algorithms and, in some cases, algorithm aversion ([30]), which generates negative emotions such as a sense of insecurity at work. Additionally, low algorithmic literacy limits these workers’ algorithmic perception and awareness. When platform workers are unable to understand or predict the underlying logic of algorithmic recommendations, they are prone to experiencing frustration and distrust. This lack of understanding or predictability not only impairs their utilization of tools and resources during problem-solving and decision-making but also increases their workload ([40]). Compared with high algorithmic literacy workers, those with low algorithmic literacy are more likely to have lower job performance and may even face platform-related penalties, leading them to perceive algorithms as a hindrance to their work. Thus, the following hypothesis is proposed:

**H3b.** 
*Algorithmic literacy negatively moderates the relationship between perceived algorithmic control and prevention-focused job crafting.*


### 2.5. The Moderated Mediation Effect

Based on H1c and H3a, platform workers with high algorithmic literacy possess better information comprehension and processing abilities, leading them to hold a positive attitude towards platform algorithmic control. They engage in promotion-focused job crafting, enhancing job autonomy through interaction with the algorithmic control system, and adopt proactive strategies to tackle various challenges and opportunities at work. This results in greater job achievement, self-efficacy, and work vitality, which increases work engagement. Thus, this study suggests that algorithmic literacy positively moderates the mediating effect of promotion-focused job crafting in the relationship between perceived algorithmic control and work engagement. Specifically, when platform workers have a high level of algorithmic literacy, promotion-focused job crafting exerts a stronger mediating effect. Conversely, when algorithmic literacy is low, the mediating effect of promotion-focused job crafting weakens.

**H4a.** 
*Algorithmic literacy positively moderates the mediating effect of promotion-focused job crafting in the relationship between perceived algorithmic control and work engagement.*


Combining H2c and H3b, platform workers with low algorithmic literacy, when perceiving algorithmic control, are prone to issues like unclear understanding of rules and low trust due to a lack of necessary algorithmic knowledge and skills. They are likely to engage in prevention-focused job crafting, taking a passive attitude in interactions with the algorithm, meeting only the basic requirements of algorithmic control to avoid platform penalties, while minimizing their time and effort investment in work. Therefore, this study suggests that algorithmic literacy negatively moderates the mediating effect of prevention-focused job crafting in the relationship between perceived algorithmic control and work engagement. Specifically, when algorithmic literacy is high, prevention-focused job crafting exhibits a weaker mediating effect, whereas when algorithmic literacy is low, this mediating effect strengthens. Based on this, the following hypothesis is proposed:

**H4b.** 
*Algorithmic literacy negatively moderates the mediating effect of prevention-focused job crafting in the relationship between perceived algorithmic control and work engagement.*


In summary, we propose the following theoretical model (see Figure 1):

## 3. Materials and Methods

### 3.1. Sampling and Data Collection

This study employed a questionnaire survey method, with the specific research participants being platform food delivery couriers and ride-hailing drivers. To mitigate severe common method bias, a multi-wave data collection strategy was employed. Prior to the formal survey, a pre-survey was conducted with gig workers to refine the wording of questionnaire items. The first section of the questionnaire included a screening question to confirm the eligibility of respondents. To reduce the psychological burden on respondents, the questionnaire explicitly emphasized that all survey content would be used exclusively for academic research purposes, with full anonymity and confidentiality guaranteed throughout the process; additionally, to enhance respondents’ understanding of “algorithmic control” and ensure their answers were more consistent with real-world situations, a definition and illustrative examples of “algorithmic control” were provided in the first section of the questionnaire.

The survey was conducted on the Chinese online questionnaire survey platform Credamo (https://www.credamo.com) and a two-wave follow-up survey was administered to the same cohort of Chinese respondents. Specifically, during data collection, two distinct questionnaires were distributed to the identical group of participants at two time points separated by a two-week interval. To facilitate subsequent data matching, respondents were required to provide the last four digits of their mobile phone numbers at the end of each questionnaire. The first-wave survey collected data on variables including platform workers’ perceived algorithmic control, promotion-focused job crafting, and prevention-focused job crafting and control variables; two weeks after the completion of the first-stage questionnaire, the research team invited the same respondents to participate in a follow-up survey, and the second questionnaire was used to gather data on work engagement and algorithmic literacy.

A total of 385 questionnaires were collected in the first stage of the survey, and 342 matched questionnaires were collected in the second stage, resulting in a response rate of 88.83% for the follow-up survey. To ensure data quality, the study screened the questionnaires based on specific criteria: questionnaires with a completion time shorter than 1 min or longer than 15 min were excluded, and questionnaires with patterned responses to multiple items or irregular answering were also eliminated. After screening, 302 valid questionnaires were finally retained, with an effective response rate of 88.30%. The demographic characteristics of the sample are shown in Table 1. In addition, 72.18% of the respondents have a job tenure of more than one year. The sample structure in which 28% have a tenure of less than one year is consistent with the actual employment situation of Chinese gig workers, particularly food delivery riders. Those with short tenures are more sensitive to the perception of algorithmic control as they are in the adaptation period, thus ensuring the reliability of the sample quality.

### 3.2. Measurement

All scales for the core variables in this study were derived from well-established instruments, among which the scales for four variables—promotion-focused job crafting, prevention-focused job crafting, algorithmic literacy, and work engagement—originated from Western contexts. To address this, we adopted the “translation-back translation” method to translate the English scales into Chinese, with adaptive adjustments made to align them with the Chinese management context. Prior to the formal survey, a pre-survey was conducted to verify the applicability of the scales in the Chinese context and ensure that all items were clearly expressed and easily understandable. Except for the control variables, all scale items were measured using a 5-point Likert scale, with responses ranging from 1 (strongly disagree) to 5 (strongly agree) to reflect increasing degrees of agreement. A total of 70 platform gig workers were recruited as respondents for the pre-survey. The analysis results showed that the Cronbach’s α coefficients for perceived algorithmic control, promotion-focused job crafting, prevention-focused job crafting, algorithmic literacy, and work engagement were 0.875, 0.922, 0.793, 0.941, and 0.925, respectively, all exceeding the acceptable threshold of 0.7. The KMO values were 0.870, 0.877, 0.781, 0.929, and 0.735, respectively, also all above the critical value of 0.7. These results indicate that the reliability and validity of the scales used in this study meet the requirements of academic standards and are applicable for the formal survey.

Perceived algorithmic control was measured using an 11-item scale developed by [25] ([25]), with sample items including “The online platform can intelligently arrange work tasks for me.”

Promotion-focused job crafting was assessed via a 15-item scale developed by [19] ([19]), with example items such as “I continuously improve my work abilities.”

Prevention-focused job crafting was adapted from the scale developed by [19] ([19]), which comprises 6 items (e.g., “I avoid work that causes me mental stress as much as possible”).

Algorithmic literacy was measured using the scale developed by [11] ([11]), which includes 16 items (e.g., “Electronic payment can be used for algorithm development and application”).

Work engagement was evaluated with a 9-item scale developed by [28] ([28]), featuring items like “At work, I feel very capable and energetic.”

Control variables: Drawing on previous literature on perceived algorithmic control and work engagement, this study included and controlled for six individual-level factors that might influence the research outcomes: gender, age, industry type, education level, years of work experience, and employment type.

The complete measurement scales for all core variables are presented in Appendix A.

### 3.3. Statistical Analysis Methods

This study utilized SPSS 29.0 and Mplus 8.3 software to conduct statistical analyses on the questionnaire data. First, we used SPSS 29.0 to test the reliability of the main scales. Second, confirmatory factor analysis (CFA) was performed via Mplus 8.3 to evaluate the validity of the scales and examine the presence of serious common method bias. Finally, to verify the effects in the theoretical model, the nonparametric Bootstrap test method recommended by [26] ([26]) was adopted. The PROCESS macro program in SPSS was invoked for analysis, with 5000 repeated resamples and a 95% bias-corrected confidence interval set. Through this approach, a systematic test was conducted on the direct effects, mediating effects, moderating effects, and moderated mediating effects of the model in sequence.

## 4. Results

### 4.1. Reliability and Validity Test

As shown in Table 2, the Cronbach’s α coefficients for perceived algorithmic control, promotion-focused job crafting, prevention-focused job crafting, algorithmic literacy, and work engagement in the formal survey were 0.895, 0.931, 0.819, 0.826, and 0.946, respectively, all exceeding the critical threshold of 0.8, indicating good reliability for each measurement scale. The composite reliability (CR) of each key variable was tested, with all values surpassing the recommended cutoff of 0.7, providing evidence for good internal consistency. The results of CFA are shown in Table 3. The fit indices for the five-factor model (χ^2^/df = 1.499, CFI = 0.906, TLI = 0.900, RMSEA = 0.041, SRMR = 0.069) were superior to those of alternative models, demonstrating that the proposed measurement model in this study exhibits a good fit to the data. This indicates that the key variables can be clearly distinguished and possess good discriminant validity.

To verify the independence of the two mediating variables, this study specifically developed a single-mediator four-factor model by consolidating promotion-focused job crafting and prevention-focused job crafting into a single factor. The fit indices for this model were as follows: χ^2^/df = 2.151, RMSEA = 0.062, CFI = 0.778, TLI = 0.770, SRMR = 0.082. Although the χ^2^/df and RMSEA indices were marginally acceptable, the key relative fit indices, CFI and TLI, fell well below the critical threshold of 0.90, indicating a poor fit between the model and the data. In contrast, the proposed five-factor model in this study (χ^2^/df = 1.499, CFI = 0.906, TLI = 0.900, RMSEA = 0.041, SRMR = 0.069) exhibited significantly better performance across all fit indices compared to the single-mediator four-factor model. The substantial improvement in CFI and TLI, particularly both exceeding the 0.90 benchmark for excellent fit, strongly supports the discriminant validity of treating promotion-focused and prevention-focused job crafting as two distinct mediator variables. This finding also reinforces the dual-pathway conceptualization of the theoretical model in this study.

Furthermore, this study constructed an additional four-factor model by merging algorithmic literacy with other related latent constructs and compared it with the theoretically proposed five-factor model. The core purpose of this comparison was to verify the discriminant validity of algorithmic literacy as an independent latent construct. The results showed that this four-factor model (χ^2^/df = 2.066, CFI = 0.793, TLI = 0.785, RMSEA = 0.060, SRMR = 0.083) was significantly inferior to the five-factor model across all key fit indices. This comparative evidence strongly confirms that algorithmic literacy is an empirically distinguishable latent construct, thereby justifying its inclusion as an independent variable in the theoretical framework.

### 4.2. Common Method Bias

Common method bias (CMB) was tested via Harman’s single-factor test (a widely used social science approach for CMB assessment). Results showed the first unrotated factor explained 43.28% of total variance. Since this is below the 50% critical threshold (a standard for ruling out severe CMB), CMB was not a major concern in this study’s data collection.

### 4.3. Descriptive Statistics and Correlation Analysis

The results of descriptive statistics and correlation analysis are shown in Table 4. After controlling for the control variables, perceived algorithmic control was positively correlated with both promotion-focused job crafting (r = 0.252, *p* < 0.01) and prevention-focused job crafting (r = 0.145, *p* < 0.05). Additionally, promotion-focused job crafting exhibited a positive correlation with work engagement (r = 0.754, *p* < 0.01), whereas prevention-focused job crafting was negatively correlated with work engagement (r = −0.249, *p* < 0.01).

### 4.4. Hypotheses Test

#### 4.4.1. Direct Effects and Mediating Effects Test

Bootstrapping tests were conducted using the PROCESS macro recommended by [26] ([26]) to examine mediating effects. Specifically, Model 4 in the PROCESS procedure was selected, with 5000 bootstrap samples and a 95% confidence interval (CI) specified. All variables—except for the moderating variable (algorithmic literacy)—were simultaneously included in the Process procedure, yielding the analysis results (see Table 5).

As shown in Table 5, perceived algorithmic control is significantly negatively correlated with work engagement (β= −0.241, 95% CI [−0.410, −0.072]), indicating that perceived algorithmic control has a direct negative link with employees’ work engagement. Further analysis identified two parallel partial mediation paths with opposing effect directions:

Perceived algorithmic control was significantly positively associated with promotion-focused job crafting (effect = 0.429, *p* < 0.01), thereby supporting H1a. In turn, promotion-focused job crafting exhibited a significant positive association with work engagement (effect = 0.488, *p* < 0.001), which supports H1b. The indirect effect of perceived algorithmic control on work engagement via promotion-focused job crafting was statistically significant (effect = 0.209, 95% CI = [0.128, 0.408]), accounting for 29.4% of the absolute value of the total effect. This finding demonstrates that perceived algorithmic control can exert an indirect positive impact on work engagement through promotion-focused job crafting, thus validating H1c.

Perceived algorithmic control was significantly positively associated with prevention-focused job crafting (effect = 0.456, *p* < 0.01), providing support for H2a. Conversely, prevention-focused job crafting was significantly negatively associated with work engagement (effect = −0.087, *p* < 0.01), which supports H2b. Meanwhile, the indirect effect of perceived algorithmic control on work engagement via prevention-focused job crafting was significant (effect = −0.039, 95% CI = [−0.086, −0.005]), constituting 54.9% of the absolute value of the total effect. This suggests that perceived algorithmic control can exert an indirect negative influence on work engagement through prevention-focused job crafting, verifying H2c.

Collectively, promotion-focused and prevention-focused job crafting both function as partial mediators in the relationship between perceived algorithmic control and work engagement. Employees’ regulatory focus tendencies shape their interpretations and responses to algorithmic control, which in turn lead to markedly divergent work engagement outcomes.

#### 4.4.2. Moderating Effects Test

This study employed the bootstrap method to test the moderation effect of the model. Specifically, we selected Model 1 in the Process procedure, with 5000 bootstrap samples and a 95% CI. The independent variable (PAC), mediating variables (PRO and PRE), moderating variable (AL), and control variables were all entered into the Process procedure to obtain the moderation effect analysis results. The results are shown in Table 6.

As shown in Model 2 (Table 6), which incorporates the interaction term between mean-centered perceived algorithmic control and algorithmic literacy, the interactive effect of perceived algorithmic control and algorithmic literacy on promotion-focused job crafting was statistically significant (β = 0.355, *p* < 0.001). This finding supports H3a, indicating that algorithmic literacy positively moderates the relationship between perceived algorithmic control and promotion-focused job crafting. To further unpack this interaction, the sample was split into high and low algorithmic literacy groups (one standard deviation above/below the mean). Simple slope analyses revealed a stronger positive relationship for platform workers with high algorithmic literacy (simple slope = 0.880) than their low literacy counterparts (simple slope = 0.465), as visualized in Figure 2.

As presented in Model 4 (Table 6), which includes the interaction term between mean-centered perceived algorithmic control and algorithmic literacy, the interaction coefficient was significant (β = −0.751, *p* < 0.05). This result supports H3b, demonstrating that algorithmic literacy negatively moderates the relationship between perceived algorithmic control and prevention-focused job crafting. To further unpack this interaction, the sample was split into high and low algorithmic literacy groups (one standard deviation above/below the mean). Simple slope analyses showed a positive slope for platform workers with low literacy (simple slope = 0.345) and a reversed negative slope for platform workers with high literacy (simple slope = −0.126), as plotted in Figure 3.

From a practical standpoint, platform workers with high algorithmic literacy can more effectively interpret algorithmic control as a clear behavioral framework and performance feedback tool, thereby stimulating their enthusiasm for proactively optimizing their work processes. Therefore, when implementing algorithmic management, enterprises should enhance platform workers’ ability to understand and apply algorithms through targeted training programs. This approach can help achieve the goal of efficiency improvement through effective human–machine collaboration.

#### 4.4.3. Moderated Mediating Effect Test

To examine whether the mediating effects vary across employees with different levels of algorithmic literacy (high vs. low), this study utilized SPSS PROCESS macro to estimate the moderated mediating effects. Specifically, a Bootstrap analysis with 5000 resamples was conducted to test the significance of the conditional indirect effects.

As shown in Path 1 of Table 7, the indirect effect of perceived algorithmic control on work engagement via promotion-focused job crafting was 0.235 (95% CI = [0.196, 0.380]) under low algorithmic literacy (mean minus one SD) and 0.446 (95% CI = [0.282, 0.667]) under high algorithmic literacy (mean plus one SD). The difference between these indirect effects was 0.341 (95% CI = [0.212, 0.482]). These findings suggest that the mediating effect of promotion-focused job crafting is stronger at high levels of algorithmic literacy. Thus, algorithmic literacy positively moderates the indirect relationship between perceived algorithmic control and work engagement through promotion-focused job crafting, supporting H4a.

As shown in Path 2 of Table 7, the indirect effect of perceived algorithmic control on work engagement via prevention-focused job crafting was −0.049 (95% CI = [−0.103, −0.101]) under low algorithmic literacy (mean minus one SD), and −0.005 (95% CI = [−0.046, 0.034]) under high algorithmic literacy (mean plus one SD). The difference between these indirect effects was −0.027 (95% CI = [−0.063, −0.002]). These results indicate that the mediating effect of prevention-focused job crafting is stronger for platform workers with low algorithmic literacy than for those with high algorithmic literacy. Therefore, algorithmic literacy negatively moderates the indirect effect of perceived algorithmic control on work engagement through prevention-focused job crafting, supporting H4b.

These findings clearly reveal that algorithmic literacy is not merely a technical competency indicator, but a key psychological resource that influences how platform workers respond to algorithmic management and ultimately shapes their work states. This implies that platform organizations can effectively guide platform workers to interpret algorithmic rules as opportunities rather than threats by enhancing platform workers’ algorithmic literacy through systematic training.

## 5. Discussion

Although algorithmic management has become the cornerstone of the platform economy, there remain significant controversies regarding its impact on platform workers’ psychology and behaviors ([4]; [23]). On the one hand, algorithms are posited to enhance efficiency and fairness; on the other hand, their opaqueness and coercive potential can trigger worker alienation and resistance ([38]; [9]). Based on regulatory focus theory, this study constructed a “double-edged sword” model to examine the impact of perceived algorithmic control on work engagement among platform workers. Specifically, promotion-focused job crafting and prevention-focused job crafting were incorporated as mediating variables, while algorithmic literacy was introduced as a moderating variable. Empirical analyses were conducted using 302 valid data samples, and the key findings are as follows:

First, perceived algorithmic control exerts a positive indirect effect on work engagement through promotion-focused job crafting, whereas it exerts a negative indirect effect on work engagement through prevention-focused job crafting. This dual-path mechanism aligns with the core tenets of regulatory focus theory, which posits that external stimuli can activate distinct motivational systems with divergent outcomes. From the perspective of regulatory focus theory, the overall effect of perceived algorithmic control on work engagement (PAC→WE) is a composite outcome of the joint action of these two mediating paths. Specifically, the mediating effect of promotion-focused job crafting is weaker than that of prevention-focused job crafting, and thus, the overall impact of perceived algorithmic control on platform workers’ work engagement is negative.

Second, algorithmic literacy positively moderates both the direct relationship between perceived algorithmic control and promotion-focused job crafting and the indirect mediating effect of promotion-focused job crafting in the link between perceived algorithmic control and work engagement. This finding aligns with research suggesting that digital literacy empowers individuals in technology-saturated environments ([11]). Platform workers with high algorithmic literacy possess stronger adaptability to the technological environment, hold a positive attitude towards platform algorithmic control, and are more inclined to engage in promotion-focused job crafting. They utilize algorithms and related technical tools to solve specific problems encountered in their personal work, and accurately align with platform demands through perceived algorithmic control to seek more benefits for their individual development, thereby increasing their work engagement.

Third, algorithmic literacy positively moderates the direct relationship between perceived algorithmic control and promotion-focused job crafting, while negatively moderating the indirect mediating effect of prevention-focused job crafting in the association between perceived algorithmic control and work engagement. This result corroborates studies on algorithm aversion and distrust, which often stem from a lack of understanding ([30]). Individuals with low algorithmic literacy, due to an unclear understanding of the platforms’ motives for implementing algorithmic control, as well as a lack of knowledge and trust in algorithmic control technologies, tend to adopt a more conservative attitude to protect their personal resources when facing algorithmic control. Consequently, they are more inclined to engage in prevention-focused job crafting, which exerts a negative impact on their work engagement.

### 5.1. Theoretical Implications

First, it broadens the theoretical scope of algorithmic management research by shifting scholarly attention from platforms’ structural control to workers’ subjective perceptual experiences. Whereas existing literature predominantly adopts an organizational perspective—focusing on how algorithmic control enhances performance and ensures compliance (e.g., [23])—this research empirically establishes the relationship between perceived algorithmic control and work engagement, thereby positioning individual subjective cognition at the core of algorithmic management mechanisms. This theoretical reorientation not only aligns with [10]’ ([10]) proposition regarding the primacy of subjective perception in shaping employee behavior but also incorporates work engagement—a critical motivational construct—into the outcome framework of algorithmic management. Consequently, it offers a new theoretical lens for understanding how algorithmic control influences individual psychological states through cognitive pathways.

Second, by integrating regulatory focus theory, this study develops a dual-pathway model that reconciles the paradoxical effects of perceived algorithmic control, thereby advancing theoretical insight into its underlying psychological mechanisms. Current scholarship remains divided regarding algorithmic control’s impact: some studies highlight its positive behavioral outcomes through mechanisms such as goal specification and instant feedback ([17]; [20]), whereas others emphasize its detrimental effects resulting from diminished autonomy ([24]; [38]). Such theoretical polarization reveals a fundamental limitation in existing explanations—the inability to account for the coexistence of dual effects. Grounded in regulatory focus theory, our research reconceptualizes these seemingly contradictory findings as parallel psychological processes, demonstrating how perceived algorithmic control activates distinct self-regulatory systems that subsequently influence work engagement. This theoretical framework transcends the conventional “either-or” paradigm and establishes an integrated mechanism through which perceived algorithmic control simultaneously generates both motivating and demotivating effects.

Third, this study elevates algorithmic literacy from a static competency to a pivotal boundary condition moderating the relationship between perceived algorithmic control and work engagement, thereby advancing the theoretical conceptualization of this construct in organizational contexts. In the digital age, the importance of algorithmic literacy has become increasingly prominent. Workers with algorithmic literacy can make more rational decisions and critically evaluate algorithmic content ([20]). Although algorithmic literacy is regarded as a core competency in the digital era, its role in the process of algorithmic management has not yet been fully elaborated from a theoretical perspective. Our findings reveal that algorithmic literacy functions as a psychological resource that regulates cognition and behavior, fundamentally shaping how individuals respond to perceived algorithmic control. Workers with high algorithmic literacy tend to interpret such control as developmental opportunities, consequently exhibiting promotion-focused responses, whereas those with low literacy are more likely to perceive it as threatening, leading to prevention-focused behaviors. This moderating role is particularly salient in the Chinese context. China is not only characterized by a rapid digital transformation but also by facing a significant urban-rural digital divide. For many platform workers, algorithmic literacy is not merely a job skill but a critical form of “algorithmic capital” that determines their ability to navigate and succeed within the highly competitive platform ecosystems that dominate the Chinese urban service economy. This insight situates algorithmic literacy within the broader “structure-agency” theoretical discourse, providing a conceptual foundation for understanding why different worker groups develop divergent responses under identical algorithmic environments.

### 5.2. Practical Implications

This study reveals that perceived algorithmic control influences work engagement through dual pathways of promotion-focused and prevention-focused job crafting, with algorithmic literacy serving as a critical moderator. Based on these empirical findings, we propose the following evidence-based recommendations for platform enterprises.

First, optimize the design of algorithmic systems to strengthen the promotion pathway. The study finds that when algorithmic control is perceived as providing job resources, it triggers promotion-focused job crafting. Therefore, platforms should: (a) enhance the accuracy and transparency of algorithmic compensation calculations to ensure that the reward system aligns with market principles and stimulates individual motivation; (b) optimize incentive mechanisms by improving methods for analyzing task difficulty and intensity; (c) strengthen the scenario-based design of platform tasks by categorizing tasks into different modules and levels, thereby increasing workers’ autonomy in order selection and enhancing their enjoyment of work.

Second, balance technical control with humanized care to mitigate the prevention pathway. Given the existence of the prevention-focused job crafting pathway, platforms should, on one hand, improve feedback mechanisms by: (a) establishing regular systems to collect workers’ experiences with algorithmic functions; (b) using workers’ problem feedback and improvement suggestions as critical references for the redesign of algorithmic systems; (c) continuously optimizing algorithmic performance through technical upgrades and experience integration to address specific issues such as inaccurate positioning and route planning. On the other hand, platforms should: (a) adopt “soft control” methods such as work philosophy design and cultural promotion to enhance the sense of work meaningfulness; (b) grant workers meaningful work autonomy to strengthen their sense of control; (c) establish support systems to enhance trust and belonging, for instance, by providing constructive feedback, real-time assistance in complex situations, and meaningful recognition of achievements, rather than relying solely on punitive monitoring.

Third, implement differentiated management strategies based on algorithmic literacy. The study provides evidence that algorithmic literacy moderates the relationship between algorithmic control and job crafting. Platform enterprises can: (a) use algorithmic technologies to conduct preliminary analyses of workers’ algorithmic literacy levels; (b) for workers with high algorithmic literacy, encourage them to unlock more platform functions and grant them appropriate autonomy to enhance their self-efficacy; (c) for workers with low algorithmic literacy, improve their understanding and mastery of algorithmic technologies and operational methods through online technical training, thereby reducing their “aversion to difficulties” in work. In the Chinese context, this differentiated approach aligns with broader national strategies for “vocational skills upgrading”. Platforms can collaborate with vocational education and training systems to provide certified algorithmic literacy courses. This not only enhances worker competency but also contributes to social stability by improving the employment competitiveness and adaptability of a vast flexible workforce, many of whom view platform work as a pathway to urban integration and upward mobility.

### 5.3. Limitations and Future Studies

First, the generalizability of this study’s findings is constrained by its specific research context. All participants in this study are Chinese gig workers; although a random sampling method was adopted, the sample is restricted to the Chinese domestic group, which still limits its representativeness. Meanwhile, shaped by the collectivist cultural value of “group cooperation” inherent in Chinese culture, platform workers’ perceptions of algorithmic control tend to prioritize alignment with group norms, rather than judging its rationality purely from an individual perspective. These sampling characteristics and cultural context jointly restrict the extrapolation of the study’s conclusions to other cultural settings, and also confine the theoretical implications of this study mainly to the landscape of China’s platform economy. Future research should adopt multi-channel sampling strategies to expand sample sources across different countries and cultural backgrounds, so as to further enhance the external validity of the research findings.

Second, although the study followed the translation-back translation procedure for scale adaptation, the study did not report confirmatory factor analysis results for cross-language validity. Furthermore, the study did not control for several potential confounding variables such as job autonomy, platform dependency, and income stability, which might influence the relationships examined. Future research should address these limitations by rigorously validating translated scales, incorporating objective measures where possible, and including relevant control variables to establish more robust causal inferences.

Third, we did not examine the structural dimensions of algorithmic control—including how platform architecture, ownership models, and regulatory frameworks shape power asymmetries in platform work. Future research should investigate these crucial aspects through multi-level approaches that connect individual experiences with broader institutional arrangements.

Fourth, this study has a limitation regarding its sample structure: among the 302 respondents, 28% had a job tenure of less than one year (4.64% with less than three months). Respondents with short tenures (particularly those in the early stages of their roles) may not have fully developed perceptions of algorithmic control, which could potentially compromise response quality. Future research could address this by including “a job tenure of at least one year” as a control variable to enhance the external validity of findings.

Lastly, our focus on delivery workers and ride-hailing drivers, while appropriate for this initial investigation, limits the scope of our findings. Algorithmic control—as an integrative management approach combining technological and platform functions—may manifest differently across various industries and platform types. Future studies should explore whether our findings generalize to other gig economy sectors such as freelance knowledge work, domestic services, or creative industries, where the nature of algorithmic control and worker–platform relationships might differ substantially.

## Figures and Tables

**Figure 1 behavsci-16-00033-f001:**
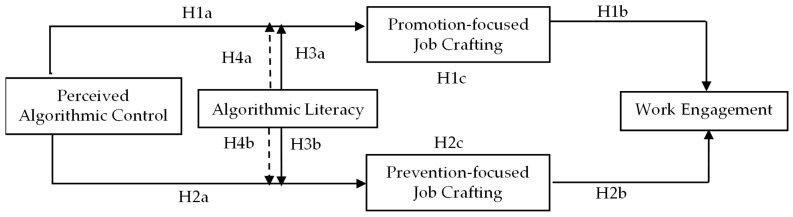
Theoretical model. Note: Dashed lines indicate the moderated mediation effect paths (H4a/H4b).

**Figure 2 behavsci-16-00033-f002:**
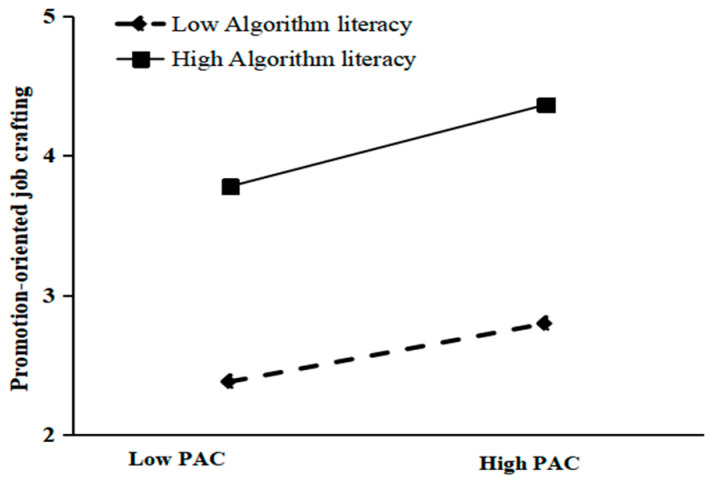
Moderating effect of AL on the relationship between PAC and PRO.

**Figure 3 behavsci-16-00033-f003:**
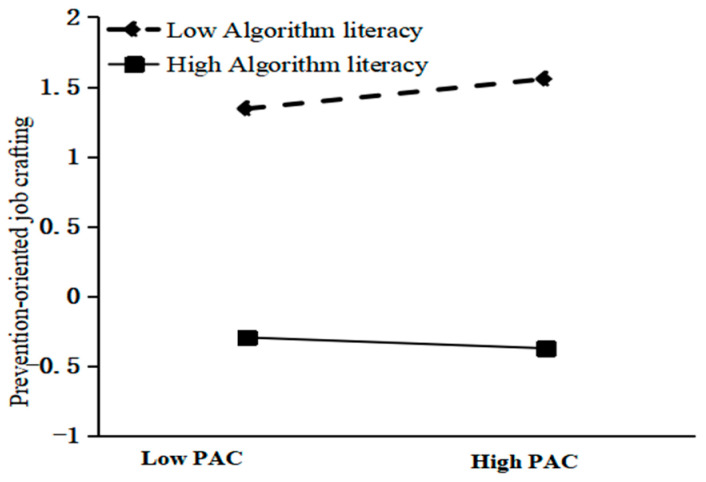
Moderating effect of AL on the relationship between PAC and PRE.

**Table 1 behavsci-16-00033-t001:** Sample description (*N* = 302).

Variable	Category	Count	Proportion	Variable	Category	Count	Proportion
Occupation	Delivery Couriers	215	71.19%	Education	Primary and Middle School	9	2.98%
	Ride-Hailing Driver	87	28.81%		High School/Vocational School	43	14.24%
Gender	Male	211	69.90%		Associate Degree	87	28.81%
	Female	91	30.10%		Bachelor’s Degree	145	48.01%
Age	18–20	18	6.00%		Master’s Degree and Above	18	5.96%
	21–40	244	80.80%	Job Tenure	≤3months	14	4.64%
	41–60	39	12.90%		3–6 months	37	12.25%
	Over 60	1	0.30%		6–12 months	33	10.93%
Employment Type	Part-time	92	30.46%		≥1 year	218	72.18%
	Full-time	210	69.54%				

**Table 2 behavsci-16-00033-t002:** Reliability and validity test.

Variable	Item	Factor Loading	Cronbach’s α	CR	AVE	Variable	Item	Factor loading	Cronbach’s α	CR	AVE
PAC	PAC1	0.657	0.895	0.918	0.505	AL	AL4	0.682	0.826	0.920	0.511
	PAC2	0.679					AL5	0.694			
	PAC3	0.697					AL6	0.808			
	PAC4	0.707					AL7	0.724			
	PAC5	0.647					AL8	0.670			
	PAC6	0.736					AL9	0.707			
	PAC7	0.721					AL10	0.752			
	PAC8	0.673					AL11	0.689			
	PAC9	0.766					AL12	0.782			
	PAC10	0.772					AL13	0.698			
	PAC11	0.750					AL14	0.831			
WE	WE1	0.729	0.946	0.951	0.640		AL15	0.746			
	WE2	0.817					AL16	0.651			
	WE3	0.845				PRO	PRO1	0.755	0.931	0.924	0.526
	WE4	0.783					PRO2	0.759			
	WE5	0.832					PRO3	0.669			
	WE6	0.843					PRO4	0.743			
	WE7	0.871					PRO5	0.705			
	WE8	0.812					PRO6	0.759			
	WE9	0.785					PRO7	0.730			
PRE	PRE1	0.769	0.819	0.856	0.500		PRO8	0.625			
	PRE2	0.683					PRO9	0.728			
	PRE3	0.682					PRO10	0.758			
	PRE4	0.717					PRO11	0.735			
	PRE5	0.744					PRO12	0.699			
	PRE6	0.638					PRO13	0.607			
AL	AL1	0.765					PRO14	0.740			
	AL2	0.689					PRO15	0.639			
	AL3	0.670									

Note(s): *N* = 302, PAC = perceived algorithmic control, WE = work engagement, PRO = promotion-focused job crafting, PRE = prevention-focused job crafting, AL = algorithmic literacy.

**Table 3 behavsci-16-00033-t003:** Results of confirmatory factor analysis (CFA).

Model	Factor	χ^2^/df	RMSEA	CFI	TLI	SRMR
Five-fact model	PAC, PRO, PRE, AL, WE	1.499	0.041	0.906	0.900	0.069
Four-fact model	PAC, PRO + PRE, AL, WE	2.151	0.062	0.778	0.770	0.082
Four-fact model	PAC, PRO, PRE, WE	2.066	0.060	0.793	0.785	0.083
Three-fact model	PAC, PRO + PRE + AL, WE	2.393	0.068	0.73	0.720	0.086
Two-fact model	PAC, PRO + PRE + AL + WE	2.999	0.081	0.614	0.600	0.105
One-fact model	PAC + PRO + PRE + AL + WE	3.274	0.087	0.559	0.540	0.108

Note(s): *N* = 302, PAC = perceived algorithmic control, WE = work engagement, PRO = promotion-focused job crafting, PRE = prevention-focused job crafting, AL= algorithmic literacy.

**Table 4 behavsci-16-00033-t004:** Descriptive statistics and correlation analysis (*N* = 302).

Variable	1	2	3	4	5	6	7	8	9	10	11
1 Gen	1										
2 Edu	0.092	1									
3 JT	−0.034	0.095	1								
4 JC	−0.130 *	−0.024	0.406 **	1							
5 OT	0.163 *	0.055	0.141 *	0.008	1						
6 Age	−0.162 **	0.009	0.054	−0.065	0.055	1					
7 PAC	−0.057	0.146 *	0.122 *	0.201 **	0.138 *	0.009	1(0.71)				
8 WE	0.003	−0.019	0.295 **	0.281 **	0.187 **	0.048	0.118 *	1(0.73)			
9 PRO	0.008	0.041	0.308 **	0.281 **	0.197 **	0.085	0.252 **	0.754 **	1(0.80)		
10 PRE	0.036	0.002	−0.072	−0.110	0.057	0.054	0.145 *	−0.249 **	−0.267	1(0.71)	
11 AI	−0.015	0.015	0.143 *	0.132 *	0.054	0.015	0.067	0.527 **	0.684 **	−0.402	1(0.71)
M	1.301	2.471	3.507	1.695	1.288	2.080	2.790	4.015	3.752	3.442	3.804
SD	0.460	0.817	0.881	0.461	0.454	0.444	0.310	0.573	0.819	0.820	0.746

Note(s): *N* = 302; ** *p* < 0.01; * *p* < 0.05; PAC = perceived algorithmic control, WE = work engagement, PRO = promotion-focused job crafting, PRE = prevention-focused job crafting, AL = algorithmic literacy. Gen = Gender, Edu = Education, JT = job tenure, JC = job category, OT = occupational type. Values in parentheses are square roots of AVE.

**Table 5 behavsci-16-00033-t005:** Results of mediating effect test.

Path	Effect	Boot S.E.	*p*	Boot LLCI	Boot ULCI
PAC → WE	−0.241	0.086	0.000	−0.410	−0.072
PAC → PRO	0.429	0.153	0.005	0.128	0.731
PAC → PRE	0.456	0.167	0.006	0.125	0.783
PRO → WE	0.488	0.034	0.000	0.421	0.554
PRE → WE	−0.087	0.031	0.005	−0.148	−0.072
PAC → PRO → WE	0.209	0.100	0.000	0.128	0.408
PAC → PRE → WE	−0.039	0.021	0.000	−0.086	−0.005

Note(s): PAC = perceived algorithmic control, WE = work engagement, PRO = promotion-focused job crafting, PRE = prevention-focused job crafting.

**Table 6 behavsci-16-00033-t006:** Results of hierarchical regression analysis.

Predictors		PRO				PRE			
	Model	Model 1		Model 2		Model 3		Model 4	
Variable		b	S.E.	b	S.E.	b	S.E.	b	S.E.
Gen		0.158	0.746	0.666	0.665	0.105	0.092	0.076	0.097
Edu		0.144	0.041	0.133	0.096	−0.017	0.053	−0.016	0.053
JT		0.114	0.042	0.179	0.178	−0.007	0.058	−0.008	0.054
JC		0.185	0.082	0.026	0.030	−0.114	0.106	−0.104	0.106
OT		0.236	0.075	0.116	0.113	0.162	0.099	0.173	0.982
Age		0.174	0.078	0.214	0.221	0.127	0.112	0.124	0.102
AL		0.473 ***	0.123	0.241 ***	0.242	0.345 ***	0.159	0.578 **	0.186
PAC × AL				0.355 ***	0.359			−0.751 **	0.282
F		3.204		3.498		6.022		3.026	
△R^2^		0.473 ***		0.001 ***		0.016 ***		0.008 ***	

Note(s): *N* = 302; *** *p* < 0.001; ** *p* < 0.01; PAC = perceived algorithmic control, PRO = promotion-focused job crafting, PRE = prevention-focused job crafting, AL = algorithmic literacy, Gen = Gender, Edu = Education, JT = job tenure, JC = job category, OT = occupational type.

**Table 7 behavsci-16-00033-t007:** Results of moderated mediating effect test.

**Path1: PAC → PRO → WE**				
Effect	Effect	Boot S.E.	Boot LLCI	Boot ULCI
Low (−1 SD)	0.235	0.900	0.196	0.380
High (+1 SD)	0.446	0.984	0.282	0.667
Index of moderated mediation	0.341	0.693	0.212	0.482
**Path2: PAC → PRE → WE**				
Effect	Effect	Boot S.E.	Boot LLCI	Boot ULCI
Low (−1 SD)	−0.049	0.023	−0.103	−0.101
High (+1 SD)	−0.005	0.019	−0.046	0.034
Index of moderated mediation	−0.027	0.015	−0.063	−0.002

## Data Availability

The raw data supporting the conclusions of this article will be made available by the authors on request.

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
