# Peer review of "The “Double-Edged Sword” Effect of Perceived Algorithmic Control on Platform Workers’ Work Engagement"

_behavsci, 2025, doi:10.3390/bs16010033_

Round 1
Reviewer 1 Report
Comments and Suggestions for Authors
Overall: This study expresses a profession setting matter that resonates with many livelihoods of platform workers, yet it is a topic that is under researched. Therefore, this manuscript is timely, and the contribution is novel and insightful to the platform workers and the platform enterprises they work for in the China context.
Introduction: There are dashes ( - ), do relook in the sentence syntax so the sentences at rows 53-54 can be appealing for readers. It is a clear theoretical contribution of cohesive dual-pathway model underpinned by regulatory focus theory (Higgins et al., 2001) which is appropriate for this study. The research aims and research objectives are clearly defined in the last paragraph of this section.
Theory and Hypotheses: The literature from past studies cover from 2002 to 2023. Consider having up-to-date references in 2024 and 2025. Figure 1 diagram needs to be rechecked as the moderating and mediating variables are not clear as well as the H4a and H4b could be reflected in the theoretical framework.
Section 3.1: Seen the 302 respondents and 28% work less than one year and 4.64% is working less than 3 months. An observation would be the platform workers with short work tenure, and if the respondents at early stages as platform workers would be providing quality responses. For this study, it is acceptable since the sample size is large with 302 usable responses. So, no further action needed this time but work tenure of at least one year could be another control variable. A suggestion for future research. There are dashes ( - ), do relook in the sentence syntax so the sentences can be appealing for readers (rows 405 to 406).
Section 3.2: Please explain if a pre-test and pilot test were conducted since the scale measurements had to undergo the translation-back translation steps.
Sections 4.1 to 4.43: The authors explained the iterations performed for the best fitting model which is a good step and the results interpretation are well explained. There are dashes ( - ), do relook in the sentence syntax so the sentences can be appealing for readers (rows 516 to 517)
Section 5: Clear discussion to loop back to the research aim and research objectives. The theoretical and practical contributions are well articulated. Agree with the suggestions for future research. There are dashes ( - ), do relook in the sentence syntax so the sentences can be appealing for readers (rows 584 to 585, 677, 684)
Reviewer 2 Report
Comments and Suggestions for Authors
The introduction is logically structured and clearly conveys the significance of the study. The authors successfully situate their work within the existing literature and articulate why the topic is both academically and practically important. On this basis, it is reasonable to expect that the findings will yield substantial theoretical contributions as well as meaningful practical implications.
One additional point the authors may consider is providing a brief description of the platform economy and labor context in China. Since the study is based on Chinese platform workers, outlining key characteristics of China’s platform industry, regulatory environment, and labor conditions would further strengthen the relevance of the study. Such contextual information would help readers better understand why the proposed relationships may manifest in this setting and would enhance the overall value and interpretability of the findings.
The theoretical background is well aligned with the objectives of the study, and the authors present the relevant concepts and prior findings in a clear and coherent manner. The rationale for each hypothesis is also well developed and logically convincing, making the proposed relationships sufficiently persuasive.
The statistical analyses are conducted appropriately and appear to be executed without any major issues. The methods are suitable for the research questions, and the results are presented in a clear and coherent manner.
In the discussion section, the authors interpret the findings in a coherent manner; however, the interpretation would be more convincing if it were more explicitly situated within the existing theoretical and empirical literature. At several points, the discussion refers to prior theories or general research trends, but specific citations are not provided. Strengthening the discussion by integrating concrete references to previous studies would enhance both the academic rigor and the interpretive depth of this section.
In the section on Theoretical Implications, some portions of the discussion appear to go beyond the scope of what this section typically covers. Several statements resemble content that is more appropriately addressed in the main Discussion, particularly where the authors interpret their findings in detail or compare them directly with prior studies. The Theoretical Implications section would be clearer and more aligned with academic conventions if it focused on summarizing the study’s theoretical contributions, while relocating the more interpretive or comparative elements to the Discussion section.
In both the theoretical and practical implications, the authors do not sufficiently address the unique organizational or cultural characteristics of Chinese platform workers. Since the data were collected exclusively from China, the implications would be stronger and more meaningful if they considered how features of the Chinese platform economy—such as algorithmic management practices, labor regulations, or cultural norms—may shape the observed relationships. Incorporating this contextual perspective would enhance the depth and relevance of the implications.
Reviewer 3 Report
Comments and Suggestions for Authors
The article is interesting and relevant, but there are aspects that need to be corrected. In order to improve the quality of the article, I recommend the following:
I recommend using consistent key terms throughout the article, e.g. used "algorithmic literacy" and "algorithm literacy".
The introduction should include an additional paragraph explaining the relationship between "perceived algorithmic control" and "algorithmic management".
The theoretical part requires a more in-depth analysis of the assumptions. The theoretical part also recommends further refining the justification for each hypothesis separately (H1a, H1b, H1c; H2a, H2b, H2c).
In the methodology section, provide more justification as to whether the sample size of 302 platform workers in China is sufficient. Are the conclusions applicable to the entire population? It is not clearly explained how the linkage between respondents in the first and second waves was ensured. More information about the sampling strategy is needed.
In the results section, it is recommended to review the values and correct any inaccuracies. For example, [The substantial 412 improvement in CFI and TLI, particularly both exceeding the 0.95 benchmark for excellent 413 fit, strongly supports the discriminant validity of treating promotion-focused and 414], the text states that elsewhere in the text and in Table 3, the CFI and TLI values are lower, e.g., CFI = 0.906, TLI = 0.900, etc.
When describing mediation, indicate whether it is "full" or "partial" and provide the overall proportion of indirect/total effect.
The text in section 5.5 repeats statistical procedures (bootstrap, 95% CI, etc.) too often, but says relatively little about the conceptual aspects, e.g., what this means in practice.
It is recommended to review the entire results section in detail, as there are discrepancies, e.g., [This result supports H3a, indicating that algorithm literacy negatively moderates the 492 relationship between perceived algorithmic control and prevention-focused job crafting. 493], when hypothesis H3a is [H3a: Algorithmic literacy positively moderates the relationship between perceived 282 algorithmic control and promotion-focused job crafting. 283]
In Table 5, the PAC→WE Effect value should be −0.241, as only this coefficient corresponds to the specified CI and p value (significant, p < 0.001). It is recommended to carefully review all tables.
In the results section, it is recommended to present a complete structural model.
The discussion lacks a more critical approach.
It is recommended to describe the limitations and future research directions in more detail.
There is a case of self-citation - Zhu, J., Zhang, B., & Wang, H. (2024). The double-edged sword effects of perceived algorithmic control on platform workers’ service 789 performance. Humanities and Social Sciences Communications, 11(1), 1-12. https://doi.org/10.1057/s41599-024-02812-0
Comments on the Quality of English Language
The English could be improved to more clearly express the research.
Reviewer 4 Report
Comments and Suggestions for Authors
The choice to mobilise Regulatory Focus Theory and to distinguish promotion- and prevention-focused job crafting is well motivated, and the inclusion of algorithmic literacy as a boundary condition is conceptually appealing. Overall, the theoretical framework is rich and the model is, in principle, coherent. That said, the central research question remains somewhat implicit. The reader can infer from the model and hypotheses that the study aims to examine how perceived algorithmic control relates to engagement via distinct forms of job crafting, depending on workers’ level of algorithmic literacy, but this is never stated explicitly as a research question or purpose. A concise sentence at the end of the introduction, spelling out the key question (or a small set of questions), would markedly improve the logical clarity of the manuscript and provide a clearer thread that links the subsequent sections. Regarding methodology, the use of a two-wave survey design is a sensible choice and goes in the right direction with respect to temporal separation of measures. The sampling strategy and context are described, and the measurement instruments are grounded in previous work. At the same time, all focal variables are self-reported and the sample is non-probabilistic and restricted to a specific national context. These limitations are acknowledged, yet the language throughout the paper often suggests strong causal “effects”. Given the design, it would be more accurate to refer to associations and indirect associations, and to be more cautious when drawing causal implications. Aligning the wording with what the data can actually support would strengthen the scientific credibility of the work. The main area that requires substantial revision concerns the presentation and interpretation of the quantitative analyses. The confirmatory factor analysis is generally appropriate, but there is a mismatch between the reported values and the way they are interpreted. The manuscript states that CFI and TLI exceed a benchmark of 0.95, when the values reported are closer to 0.90. This needs to be corrected, and the authors should reconsider the benchmark they are using and the way they justify the adequacy of model fit. Similarly, the comparison between the five-factor and four-factor models is described as “confirming” the necessity of including the moderation effect of algorithmic literacy. A CFA comparison can show that five latent constructs are empirically distinguishable; it does not test moderation as such. The moderation is tested later in the PROCESS analyses, and the text should distinguish clearly between these different steps. The tables reporting mediation and moderated-mediation analyses also contain internal inconsistencies that must be resolved. In one table, the direct path from perceived algorithmic control to work engagement is reported as a positive coefficient, yet the associated confidence interval is entirely negative. In another, the bounds of a bootstrap confidence interval appear to be reversely ordered, with the upper bound smaller than the lower bound. These kinds of discrepancies suggest either transcription errors or confusion in interpreting the software output. They are relatively easy to fix, but at present they weaken the perceived soundness of the results. I would strongly encourage the authors to revisit their analyses, re-run the models as needed, and carefully cross-check every coefficient, standard error and confidence interval in the tables. In the moderation section, the narrative sometimes mixes up the two hypotheses concerning promotion- and prevention-focused job crafting. There is, for instance, a passage where a result for the promotion-focused path is said to support the hypothesis formulated for the prevention-focused path. This is likely a simple wording oversight, yet in a hypothesis-driven paper precision in this respect is important. A systematic check to ensure that each hypothesis is consistently labelled and interpreted throughout the results and discussion would be beneficial. The discussion succeeds in connecting the empirical findings back to the theoretical framework and draws out some interesting implications, notably the idea that algorithmic control can be framed either as an opportunity structure or as a source of threat, depending on workers’ regulatory focus and literacy. However, at times the tone becomes overly strong in relation to what the data can support, suggesting that the study “confirms” or “demonstrates” broad claims about algorithmic management. Given the cross-sectional nature of most of the data and the specific context of the sample, moderating these claims and framing them as contributions to an ongoing debate, rather than definitive answers, would make the argument more convincing. From a writing perspective, the manuscript is generally readable and the English is functional, but there are places where clarity could be improved. Tightening some long sentences, ensuring consistent terminology (for example, when referring to the two types of job crafting and to the different components of regulatory focus), and correcting a few slips where hypotheses or variables are mis-named would significantly enhance the overall flow.
Comments on the Quality of English LanguageThe manuscript is generally readable and the English is functional, but the expression can be improved to more clearly convey the arguments and results. Several sentences are overly long or complex, some terminology is used inconsistently (especially in relation to the two forms of job crafting and the hypotheses), and there are a few slips where variables or hypotheses are mis-labelled. A careful language edit by a proficient English speaker or professional editor would help to tighten the prose, ensure consistent wording and enhance overall clarity.
Round 2
Reviewer 3 Report
Comments and Suggestions for Authors
The authors took the comments into account and revised the article. Several further revisions are recommended:
Explain the PAC → WE effect more clearly. It is recommended to write 1-2 sentences in the discussion that would link the result and the theory.
Strengthen the limitations by emphasizing the Chinese context. It is recommended to add 1-2 sentences about the cultural context.
Comments on the Quality of English LanguageThe English could be improved to more clearly express the research.
